# scMoE: single-cell Multi-Modal Multi-Task Learning via Sparse Mixture-of-Experts

## Abstract

Recent advances in measuring high-dimensional modalities, including protein levels and DNA accessibility, at the single-cell level have prompted the need for frameworks capable of handling multi-omics data while addressing multiple tasks. Despite these advancements, most work remains limited, often focusing on either a single-modal or single-task perspective. A few recent studies have ventured into multi-omics and multi-task learning, but we identified a ① **Optimization Conflict** issue, leading to suboptimal results when integrating additional modalities in the single-cell domain. Furthermore, there is a ② **Costly Interpretability** challenge, as current approaches largely rely on costly post-hoc methods like SHAP. Motivated by these challenges, we introduce scMoE[1], a novel framework that applies Sparse Mixture-of-Experts (SMoE) within the single-cell domain. This is achieved by incorporating an SMoE layer into a transformer block with a cross-attention module. Thanks to its design, scMoE inherently provides mechanistic interpretability, a critical aspect for understanding underlying mechanisms in biological data. Furthermore, from a post-hoc perspective, we enhance interpretability by extending the concept of activation vectors (CAVs) to the single-cell domain. Extensive experiments on simulated dataset, `Dyngen`, and real-world multi-omics single-cell datasets, including {`DBiT-seq`, `Patch-seq`, `ATAC+gene`}, demonstrate the effectiveness of scMoE.

## 1 Introduction

Given the inherently multi-modal nature of multi-omics, which includes transcriptome, genome, and proteome data at the single-cell level (Lee et al., 2020), there exists a notable mismatch with current methodologies. These methods are predominantly tailored for single-modality applications, targeting specific tasks (Van Dijk et al., 2018; Yun et al., 2023; Xiong et al., 2019; Cheung et al., 2021), thereby limiting their generalizability in a multi-modal environment encompassing diverse tasks. Such tasks encompass the identification of joint groups, such as cell type across different modalities, and cross-modal prediction, where one modality is utilized to infer the expression of cells in another. Recently, UnitedNet (Tang et al., 2023) proposed a multi-task learning framework given its multi-modal nature, employing an encoder-common fuser-decoder framework based on a shared latent space. However, this approach encounters two fundamental limitations:

① **Optimization Conflict.** As illustrated in Figure 1 (a), despite UnitedNet's capability in handling a diverse multi-modal environment, its peak performance is achieved using a subset of modalities (specifically, pre-MRNA and mRNA), rather than all four modalities, which paradoxically show the worst performance among the variations. This counterintuitive outcome, given the amount of information involved, is undesirable and highlights a limitation in harnessing the full potential of multi-modal data in the single-cell domain. A deeper investigation, as depicted in Figure 1 (b), reveals that the core issue stems from the encoder-common fuser-decoder framework in multi-task settings, leading to an optimization conflict. This conflict arises because the fuser consolidates information from each modality into a shared parameter space responsible for handling different tasks. This observation underscores the need for a framework designed to disentangle the parameter space, allowing for the coexistence of both common and specialized knowledge tailored for diverse tasks.

---

[1] single-cell Sparse Mixture-of-Experts. Source code can be found in Supplementary files.

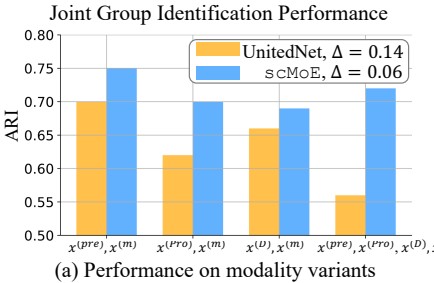 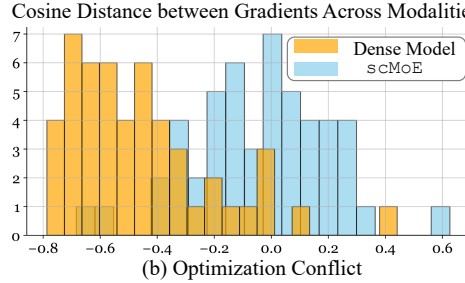

(a) Performance on modality variants  (b) Optimization Conflict

Figure 1: (a) Joint group identification across modality variants shows that UnitedNet, when using all modalities (Protein ($Pro$), mRNA ($m$), pre-mRNA ($pre$), DNA ($D$)), performs the worst, with a large gap ($\Delta = 0.14$) compared to our model ($\Delta = 0.06$), which maintains stable performance across diverse modality combinations. (b) This gap arises from optimization conflicts, particularly gradient conflicts between modalities like "Protein" and "DNA." Gradients are obtained from experts and dense MLPs in scMoE and Dense Model, respectively. Unlike Dense Models such as UnitedNet, our sparse model reduces conflict, as indicated by more positive cosine distances, enabling stronger multi-omics integration. Experiments use the Dyngen dataset.

② **Costly Interpretability.** Interpretability in the biomedical and bioinformatics domain is essential, particularly for the practical application of these fields in clinical settings (Han & Liu, 2021; Karim et al., 2023). For instance, in predicting patient responses to cancer treatments using machine learning models, clinicians might hesitate to trust a model's recommendations if they lack interpretability. A model that can elucidate the genetic markers or pathways influencing its predictions enables practitioners to make more informed, potentially life-saving decisions. Although UnitedNet provides some level of interpretability through post-hoc analysis with the SHapley Additive exPlanations (SHAP) algorithm (Lundberg & Lee, 2017), this method has its drawbacks. It comes with a high computational cost due to its need to evaluate all possible combinations of features, a complexity that increases exponentially with the number of features. This poses a significant challenge in the bio domain, where providing relevant explanations in a timely manner is crucial. This brings the necessity for mechanistic interpretability, which is inherently integrated into the model's architecture, offering immediate insights during inference. Additionally, a more lightweight design for post-hoc analysis that aligns with the needs of the bio domain is also vital.

⋆ **Sparse MoE as a Solution.** SMoE (Shazeer et al., 2017a) is an advanced neural network architecture that stands out for its ability to process complex, high-dimensional data efficiently. It achieves this by dynamically selecting a subset of specialized models, i.e., experts, for each input, offering a tailored approach in terms of a data-driven approach. Here, targeting challenges of single-cell multi-modal data, we propose scMoE, which simply replaces the MLP layer in a transformer architecture with an SMoE layer so as to address the above limitations effectively. Specifically, it addresses the ① **Optimization Conflict** by employing multiple experts, thereby naturally disentangling the parameter space. This disentanglement allows for the attainment of both shared and unique knowledge tailored for each task and modality. Regarding the challenge of ② **Costly Interpretability**, scMoE addresses this by incorporating a gating network, or router, which automatically activates specific experts. This mechanism significantly enhances the model's interpretability by immediately identifying which experts are most relevant for the task at hand during inference. Moreover, the integration of a cross-attention module before the SMoE layer, based on transformer architecture (Fedus et al., 2022), further enriches interpretability. This module adeptly captures the importance of feature combinations from different modalities, facilitating solving downstream tasks with improved efficiency and insights.

In summary, our contributions are three-fold:

- We adopt the SMoE in single-cell multi-omics multi-task learning to effectively tackle both optimization conflict and costly interpretability issues.

- To enhance interpretability efficiently, we investigate the use of concept-activation vectors (CAVs), which are particularly suitable for the single-cell domain.

- We demonstrate the effectiveness of scMoE across diverse multi-omics single-cell datasets. This includes the simulations dataset Dyngen, as well as real-world datasets such as {DBiT-seq, Patch-seq, ATAC+gene}, in joint group identification and cross-modal prediction tasks.

## 2 RELATED WORK

**Multi-Modal Multi-Task learning in single-cell data.** Multi-modal learning (Makadia et al., 2008; Weston et al., 2011; Antol et al., 2015; Ramesh et al., 2022; Saharia et al., 2022; Yang et al., 2016; Dai et al., 2022; Jaegle et al., 2021) and multi-task learning (Xue et al., 2007; Hashimoto et al., 2017; Fan et al., 2022; Chen et al., 2023) have been subjects of extensive research over the years, with significant contributions from various fields. Such advancements inspired single-cell domain where uni-modal targeting signle-task, e.g., transcriptome with imputation task (Li & Li, 2018; Van Dijk et al., 2018; Wang et al., 2021; Yun et al., 2023) or clustering task (Tian et al., 2019; Lee et al., 2023), was predominant. For instance, MOFA (Argelaguet et al., 2020) disentangs variation in single-cell studies integrating different omics data types, like genomics and proteomics. totalVI (Gayoso et al., 2021), on the other hand, specifically integrates single-cell RNA sequencing data and protein abundance for a comprehensive cellular profile. WNN (Hao et al., 2021) combines single-cell RNA and protein data, creating a unified representation of cell states. Schema (Singh et al., 2021) integrates diverse single-cell omics data, including transcriptomics and electrophysiology, providing a holistic view of cellular function and state. Recently, UnitedNet (Tang et al., 2023) has been introduced targeting multi-tasks like joint group identification and cross-modal prediction by utilizing a shared-latent space in a post-hoc explainable manner. However, as mentioned earlier, it encounters an optimization conflict issue. Involving more modalities can significantly degrade overall performance while also imposing a substantial burden on post-hoc interpretability.

**Sparse Mixture-of-Experts (SMoE).** SMoE model builds on the traditional Mixture-of-Experts (MoE) framework by introducing sparsity to enhance both computational efficiency and model scalability (Shazeer et al., 2017a; Chen et al., 1999). Unlike dense models, SMoE activates only a subset of relevant experts for each task, which reduces computational load and enables it to handle complex, high-dimensional data effectively. This selective activation has proven advantageous across diverse domains, including vision and language tasks, where it can dynamically adapt different parts of the network to specialized sub-tasks or data types (Riquelme et al., 2021; Lepikhin et al., 2021; Yun et al., 2024). Although some works (Kopf et al., 2021; Minoura et al., 2021; Liu et al., 2022) have attempted to adopt the MoE design in the single-cell domain, these approaches are limited to a single-modal setting and are often *not sparsely activated; instead, they activate all experts, following a weighted sum approach.* In this work, we focus on the Sparse MoE perspective within a multi-modal, multi-task scenario, where experts are sparsely activated based on diverse single-cell multi-omics.

## 3 METHOD

### 3.1 PRELIMINARIES

**Joint Group Identification with Cross-Modal Prediction.** Given a multi-modal single-cell data, we aim to solve a multi-task problem. The first task, joint group identification, is to identify jointly expressed characteristics, a commonality across cells despite their differing modalities such as cell type, states, or tissue regions. From a classification perspective, both unsupervised and supervised approaches can be utilized simply by modifying the loss function. Simultaneously, we aim to address the cross-modal prediction task, which infers the information, i.e., expression of cells in one modality, using data from other modalities. Considering technical noise or properties that are difficult to measure, such predictions have the potential to significantly impact the real-world single-cell domain.

**SMoE Notation.** To address the optimization conflict issue identified previously, we implement SMoE to separate the parameter space across different tasks and modalities. In our model architecture, which is based on the transformer block, we substitute the traditional feedforward network with an SMoE layer, as depicted in Figure 2(b). Formally, the SMoE comprises several experts, denoted as $f_1, f_2, \ldots, f_E$, where E represents the total number of experts, and a routing mechanism, $\mathcal{R}$, which selects experts in a sparse fashion. For a given embedding $\mathbf{x}$, the top-$k$ experts are engaged by $\mathcal{R}$ based on the highest scores $\mathcal{R}(\mathbf{x})_i$, where $i$ indicates the expert index. This procedure is expressed as follows:

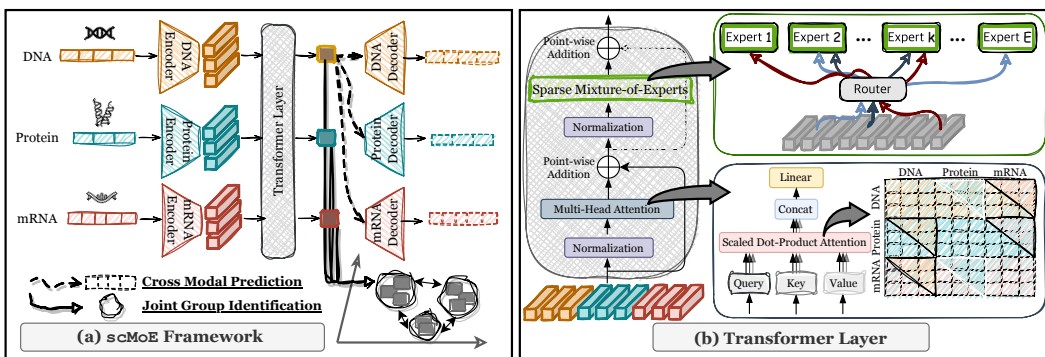

Figure 2: (a) The overview of scMoE: In the single-cell domain, each modality undergoes processing by its specific encoder, integrates with a shared transformer layer, and then passes through a modality-specific decoder. This structure enables simultaneous cross-modal prediction and group identification directly from the transformer output. (b) The transformer layer employs multi-head attention on concatenated modalities' inputs, facilitating intra-modality self-attention and inter-modality cross-attention. A Sparse Mixture-of-Experts Layer then supersedes the standard feed-forward network, enhancing the model's efficacy in single-cell multi-modal multi-task learning.

$$
\begin{aligned}
\mathbf{y} &= \sum_{i=1}^{k} \mathcal{R}(\mathbf{x})_i \cdot f_i(\mathbf{x}), \\
\mathcal{R}(\mathbf{x}) &= \text{Top-K}(\text{softmax}(g(\mathbf{x})), k), \\
\text{TopK}(\mathbf{v}, k) &= \begin{cases} \mathbf{v}, & \text{if } \mathbf{v} \text{ is in the top } k, \\ 0, & \text{otherwise.} \end{cases}
\end{aligned} \tag{1}
$$

where $\mathbf{y}$, the final output of the SMoE layer, is a weighted sum of the expert representations $f_i(\mathbf{x})$ and their corresponding weights $\mathcal{R}(\mathbf{x})_i$, as determined by the router $\mathcal{R}$. Here, $g$ denotes a trainable network, typically a small Feedforward Neural Network (FNN) that ranges from one to a few layers (Shazeer et al., 2017b; Riquelme et al., 2021). The Top-K$(\cdot)$ operation selectively retains a vector $v$ if its probability is among the top $K$ probabilities; otherwise, it sets the vector to zero.

### 3.2 scMoE: single-cell MEETS SMoE

In essence, as illustrated in Figure 2, scMoE adopts the **Encoder-Transformer-Decoder** framework. Below, we detail each module accordingly.

**Encoder.** Given the multimodal nature originating from diverse environments, such as multi-omics, we initially employ modality-specific encoders, denoted as $\mathcal{E}^{(1)}, \cdots, \mathcal{E}^{(\mathcal{V})}$, where $\mathcal{V}$ represents the total number of modalities, to effectively generate informative embeddings for each modality. Notably, our use of transformer blocks (Fedus et al., 2022; Hu & Singh, 2021) requires input tokenization, differing from our matrix-formatted single-cell domain inputs where rows represent cells and columns denote specific modalities (e.g., genes or proteins), some of which, like highly variable genes (HVG) for the gene modality, vary in number. To achieve a consistent embedding shape across different modalities through tokenization, we adopt the patching method widely used in ViT-based works (Dosovitskiy et al., 2021). Thus, for an input $\mathbf{x}^{(\nu)} \in \mathbb{R}^{B \times |\nu|}$, with batch size $B$ and size of a specific modality $|\nu|$, the output after processing by $\mathcal{E}^{(\nu)}$ is the embedding $\mathbf{h}^{(\nu)} \in \mathbb{R}^{B \times P \times D}$, where $P$ and $D$ denote the desired number of patches and the hidden dimension, respectively. With these tokenized embeddings from each modality, we proceed to the transformer block, the core component of this work.

**Transformer.** The transformer block, depicted in Figure 2 (b), primarily functions as a feature extractor and includes two key components: (1) Multi-Head Attention module facilitates both intra-modal and inter-modal attention through the modality-wise concatenated tokenized embeddings, $\mathbf{h} \in \mathbb{R}^{B \times P\mathcal{V} \times D}$. This setup enables the capturing of similarities between queries and keys across

all modality combinations, with a total of $\mathcal{V}^2$. Representing the similarities in a matrix, diagonal elements would indicate self-attention within a modality (e.g., protein-protein) while off-diagonal elements signify cross-attention between different modalities (e.g., protein-mRNA), fostering a more thorough understanding of modalities. (2) The Sparse Mixture-of-Experts (SMoE) plays a crucial role in addressing optimization conflicts, thereby enhancing multi-task, multimodal environments as illustrated in Figure 1. By replacing the conventional FNN layers, SMoE enables the training of multi-experts who share common knowledge within modalities while retaining specialized knowledge in specific modalities or tasks. This is particularly pertinent in complex environments like the single-cell domain, where efficiency and specificity are essential. The output embedding from the transformer layer serves as a primary input for the unsupervised clustering loss, Deep Divergence-based Clustering (DDC) (Kampffmeyer et al., 2019), a method proven to enhance clustering in unsupervised settings. Notably, the DDC loss can be substituted with Cross-Entropy (CE) loss for supervised applications.

**Decoder.** In the single-cell domain, which often encounters noisy inputs due to dropout events (Hicks et al., 2018) and batch effects (Shaham et al., 2017), and lacks explicit supervision signals such as cell types, attaching decoder losses is a strategy to reconstruct the originally given input matrix effectively. Building on the final embeddings from the transformer block, we incorporate a total of $\mathcal{V}$ decoders, $\mathcal{D}^{(1)}, \cdots, \mathcal{D}^{(\mathcal{V})}$, meaning there is a decoder for each modality, similar to our approach with the encoder. Given our focus on the cross-modal prediction task—predicting the expression of one modality from another—and considering that each modality serves as both input to itself and to other modalities, we aggregate a total of $\mathcal{V}^2$ reconstruction losses.

**Training Procedure.** Facing a multi-task learning scenario, we aggregate two primary losses: the DDC loss (or CE loss in supervised contexts) and the Reconstruction loss, addressing joint group identification and cross-modal prediction tasks, respectively. Unlike the iterative loss update strategy employed in UnitedNet (Tang et al., 2023), our method is straightforward, enhancing adaptability for future expansions to additional modalities. The comprehensive algorithm for training scMoE is detailed in Algorithm 1.

---

**Algorithm 1** The overall procedure of scMoE.

1: **Input:** Cell matrices, $\mathbf{x}^{(\nu)}$, Encoders, $\mathcal{E}^{(\nu)}$, Decoders $\mathcal{D}^{(\nu)}$, $\forall \nu \leq \mathcal{V}$, with Transformer Layer containing MHA and SMoE
2: **Output:** Joint Group Identification, Cross-Modal Prediction
3: /* Encoder */
4: **for** $\nu = 1, \cdots, \mathcal{V}$ **do**
5:    $\mathbf{h}^{(\nu)} \leftarrow \mathcal{E}^{(\nu)}(\mathbf{x}^{(\nu)})$
6: **end for**
7: $\mathbf{h} \leftarrow [\mathbf{h}^{(1)}||\cdots||\mathbf{h}^{(\mathcal{V})}]$
8: /* Transformer */
9: $\mathbf{h}' \leftarrow \text{MHA}(\text{Norm}(\mathbf{h})) + \text{Norm}(\mathbf{h})$
10: $\tilde{\mathbf{h}} \leftarrow \textbf{SMoE}(\text{Norm}(\mathbf{h}')) + \text{Norm}(\mathbf{h}')$            ▷ Equation (1)
11: $\mathcal{L}_{\text{DDC}} = \text{DDC}(\tilde{\mathbf{h}})$
12: /* Decoder */
13: **for** $\nu = 1, \cdots, \mathcal{V}$ **do**
14:    **for** $\mu = 1, \cdots, \mathcal{V}$ **do**
15:       $\hat{\mathbf{x}^{(\nu)}} \leftarrow \mathcal{D}^{(\mu)}(\tilde{\mathbf{h}}^{(\mu)})$
16:       $\mathcal{L}_{\text{Recon}} \leftarrow \mathcal{L}_{\text{Recon}} + \text{Recon}(\mathbf{x}^{(\nu)}, \hat{\mathbf{x}^{(\nu)}})$
17: **end for**
18: **end for**
19: $\mathcal{L}_{\text{Final}} \leftarrow \mathcal{L}_{\text{DDC}} + \mathcal{L}_{\text{Recon}}$
   =0

---

## 3.3 INTERPRETABILITY OF scMoE

In the field of biology, particularly in single-cell analysis, the interpretability of a proposed model is paramount. We explore the interpretability of scMoE from two perspectives: Mechanistic and Post-hoc.

**Mechanistic Interpretability.** Mechanistic interpretability (Wang et al., 2022; Kästner & Crook, 2023) involves understanding a model's internal mechanisms and how they contribute to its decisions, observable during inference without additional training. While the integration of the SMoE layer might seem to obscure the model's interpretability, the preceding multi-head attention mechanism, which captures both intra and inter-modality significance, maintains a level of interpretability. Furthermore, the SMoE layer's gating network, or router [2], which decides which experts to activate for a given modality or task, allows for mechanistic interpretability through its data-driven decision-making process. This aspect will be demonstrated in the subsequent experimental section.

**Post-hoc Interpretability.** The mechanistic approach, while valuable, may not always be immediately transparent, as decisions such as expert selection are based on learned patterns. To complement the model's complex inner workings without delving into intricate details, the post-hoc approach (Zhang & Zhu, 2018; Zou et al., 2023) provides insights at the input level. While SHAP (Lundberg & Lee, 2017) has been recently applied in single-cell multimodal analysis, its complexity prompts us to propose a more lightweight, directly applicable interpretability method based on Concept Activation Vectors (CAV) (Kim et al., 2018b), tailored for the single-cell domain. Further details will be presented in Section 4.5.

## 4 EXPERIMENTS

### 4.1 EXPERIMENTAL SETTINGS

**Datasets.** To evaluate our method, following (Tang et al., 2023), we conducted experiments on four different datasets, including one simulated dataset and three real-world datasets. The simulated dataset is the Dyngen dataset, which contains DNA, pre-mRNA, mRNA, and protein modalities. The real-world datasets include the Patch-seq GABAergic neuron dataset (Patch-Seq dataset) with morphological, electrophysiological, and transcriptomic modalities, the Multiome ATAC + gene expression BMMCs dataset with ATAC-seq and RNA-seq modalities, and the DBiT-seq embryo dataset (DBiT-seq dataset) with mRNA, protein, and niche mRNA modalities. Each dataset contains multimodal features relevant to our study, and its statistics are provided in Table 1. For details on preprocessing and normalization techniques, please refer to Appendix B.

Table 1: Multimodal Dataset Statistics

|  | # Train | # Test | Modalities | # Features in each Modality |
|---|---|---|---|---|
| Dyngen | 400 | 100 | DNA, pre-mRNA,mRNA,Protein | [100, 100, 100, 100] |
| Patch-seq | 404 | 44 | Morphological, Electrophysiological, Transcriptomic | [1252, 68, 514] |
| DBiT-seq | 748 | 188 | mRNA, Protein, niche mRNA | [568, 22, 568] |
| ATAC+gene | 63,025 | 6,224 | ATAC-seq, RNA-seq | [13,634, 4,000] |

**Compared Methods.** To demonstrate the superior performance of scMoE on the particularly challenging joint group identification task, we compare it with 5 state-of-the-art (SOTA) multi-modal integration methods: UnitedNet (Tang et al., 2023), the weighted Nearest Neighbor (WNN) (Hao et al., 2021), Schema (Singh et al., 2021), Multi-Omic Factor Analysis (MOFA) (Argelaguet et al., 2020), and totalVI (Gayoso et al., 2021). Additionally, we introduce an "Identification only" baseline for a more exhaustive comparison, which focuses solely on the identification aspect of the task without the complexity of integrating multiple modalities. Subsequently, we benchmark the cross-modal prediction performance of scMoE against three carefully selected baselines: UnitedNet, WNN, and the aforementioned "Identification only" method. We used the hyperparameter setting proposed in their paper, and for our best setting, please refer to Appendix C.

### 4.2 SIMULATION STUDY

As shown in Table 2, when targeting the unsupervised joint group identification task, our proposed method, scMoE, achieves the best scores across various combinations of modalities. Using solely pre-mRNA, Protein, DNA, and mRNA is more beneficial than using two modalities, such as DNA

---

[2]In this paper, we adhere to a single router shared across modalities to improve interpretability when analyzing experts.

Table 2: Joint group identification task measured by ARI in simulated Dyngen dataset upon modality combinations. The performance is averaged upon 5-fold cross-validation sets.

| | Dyngen | | | |
|---|---|---|---|---|
| | Modality Combiations | | | |
| | Pre, m | Pro, m | D, m | Pre, Pro, D, m |
| scMoE | **0.75** | **0.70** | **0.69** | **0.72** |
| UnitedNet | 0.70 | 0.62 | 0.66 | 0.56 |
| Idenfication only | 0.49 | 0.41 | 0.44 | 0.50 |
| WNN | 0.43 | 0.46 | 0.43 | 0.46 |
| Schema | 0.55 | 0.57 | 0.56 | 0.56 |
| MOFA | 0.02 | 0.05 | 0.66 | 0.05 |
| totalVI | 0.07 | 0.02 | 0.02 | - |

Table 3: Cross-modal prediction task measured by $R^2$ in simulated Dyngen dataset upon modality combinations. The performance is averaged upon 5-fold cross-validation sets.

| | Dyngen | | | |
|---|---|---|---|---|
| | Modality Combiations | | | |
| | Pre, m | Pro, m | D, m | Pre, Pro, D, m |
| scMoE | **0.41** | **0.90** | **0.67** | **0.62** |
| UnitedNet | 0.39 | 0.60 | 0.52 | 0.47 |
| Idenfication only | 0.20 | 0.28 | 0.25 | 0.35 |
| WNN | 0.21 | 0.26 | 0.22 | 0.36 |

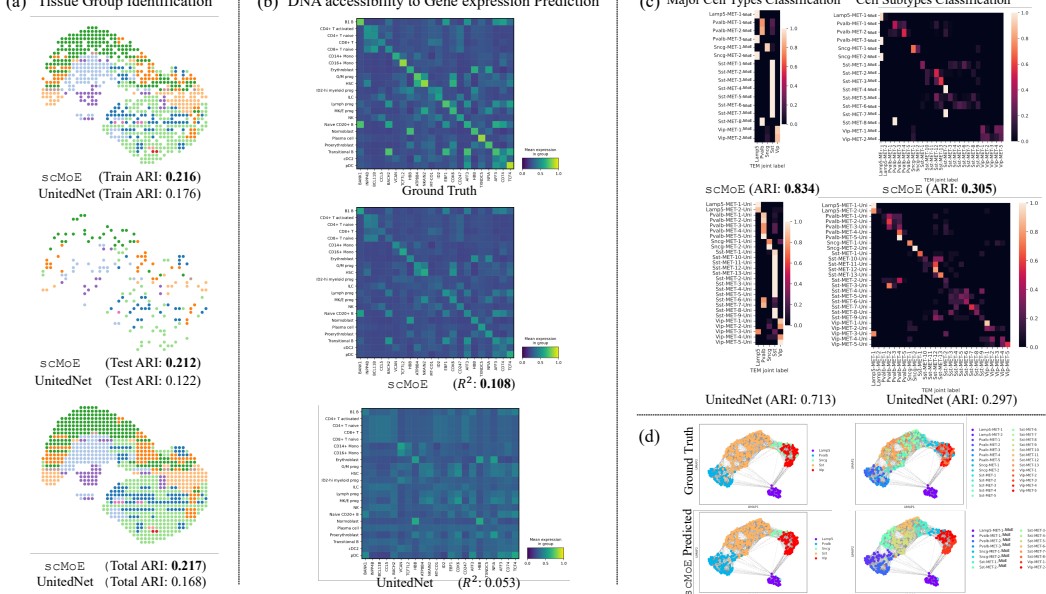

Figure 3: Performance analysis on real-world single-cell multimodal data: (a) Unsupervised tissue group identification task in the DBiT-Seq dataset. (b) Supervised cross-modal prediction task in the ATAC+gene expression BMMCs dataset. (c) Confusion matrix in Patch-seq dataset. (d) UMAP representation of latent features colored by joint cell types in Patch-seq dataset.

and mRNA. This finding contrasts with the recently proposed UnitedNet, which tends to fall short when incorporating more modalities. As corroborated by the optimization conflict issue illustrated in Figure 1, scMoE benefits from a performance gain when adding more modalities. Furthermore, in Table 3, which focuses on the cross-modal prediction task, scMoE consistently outperforms the baselines, demonstrating its effectiveness and suitability for multi-modal and multi-task learning.

## 4.3 REAL-WORLD APPLICATIONS

In this section, we compare the scMoE with recent SOTA, i.e., UnitedNet, across various real-world datasets, including DBiT-seq, ATAC+gene, and Patch-seq. In Figure 3, we have following observations: **1)** In Figure 3 (a), we perform an unsupervised tissue group identification task to verify whether scMoE appropriately clusters 13 different groups. Compared to UnitedNet, our proposed model not only performs well in the training scenario but also in the unseen test scenario, verifying its generalizability. **2)** In Figure 3 (b), using the ATAC+gene expression BMMCs dataset, we conducted a supervised cross-modal prediction task to infer gene expression given DNA accessi-

bility. Sharing a common trend in the representation of each cell type (row), thanks to the supervised signal, scMoE exhibits a similar gene expression trend in each cell type compared to the Ground Truth, outperforming UnitedNet. **3)** Utilizing the Patch-seq dataset, we create a confusion matrix comparing the joint majority cell types and cell subtypes between reference labels and each model's identified label ('-MoE' for scMoE and '-Uni' for UnitedNet), as shown in Figure 3 (c). Unlike UnitedNet, where uncertain predictions hamper overall performance, scMoE shows a notable performance gain, especially in major cell-type classification. This improvement is attributed to the gating network in the SMoE layer, making the decision process both efficient and effective. This is further supported by the UMAP representation between Ground Truth and that of scMoE, showing its effectiveness in capturing relevant cell-type specific information between similar cells.

In conclusion, scMoE demonstrates significant generalizability across multiple multi-modal single-cell datasets, excelling in tasks such as unsupervised clustering and cross-modal prediction.

## 4.4 ABLATION STUDY

To elucidate the contribution of the SMoE design within scMoE, we first conduct comprehensive ablation studies in terms of module in Table 4.4, specifically targeting the joint group identification task in the Dyngen dataset. Observations are as follows: **1)** With the dense model, possessing the same number of parameters, we observe scMoE's performance gain, showing positive effects of SMoE on alleviating gradient conflict. **2)** With the number of different experts ($N$), we observe that the current setting of 16 experts appears to be a sweet spot for SMoE configuration, as increasing the number of experts hinders their ability to specialize. **3)** For top-$k$, using top-2 is beneficial given 16 experts as a budget. **4)** We also examined the router-specific design where each modality possesses its own specific router; however, the performance rather drops, mainly because a single router equips more unified and general knowledge to handle multiple modalities simultaneously. **5)** We also varied the number of patches, which reduces the number of features in each modality-specific encoder, as used in ViT (Dosovitskiy et al., 2021), and observe that using four patches across the modalities appears to be a sweet spot, while patch numbers beyond or below this threshold lead to diminished performance.

Table 4: Ablation study of Module. Among five folds, first fold is used for the experiment. ARI used.

| | Dyngen | | | |
| --- | --- | --- | --- | --- |
| | Modality Combiations | | | |
| | Pre, m | Pro, m | D, m | Pre, Pro, D, m |
| scMoE | **0.97** | **0.83** | **0.89** | **0.75** |
| Dense | 0.68 | 0.68 | 0.71 | 0.72 |
| $N = 4$ | 0.82 | 0.71 | 0.77 | 0.75 |
| $N = 8$ | 0.80 | 0.69 | 0.76 | 0.75 |
| $N = 32$ | 0.74 | 0.65 | 0.76 | 0.74 |
| $k = 1$ | 0.79 | 0.68 | 0.73 | 0.70 |
| $k = 4$ | 0.82 | 0.70 | 0.79 | 0.74 |
| router per modality | 0.84 | 0.75 | 0.82 | 0.75 |
| 2 Patches per modality | 0.72 | 0.72 | 0.63 | 0.75 |
| 8 Patches per modality | 0.83 | 0.75 | 0.79 | 0.74 |
| 16 Patches per modality | 0.56 | 0.68 | 0.58 | 0.64 |

Table 5: Ablation study of Loss. Among five folds, first fold is used for the experiment. Dyngen dataset with four modalities, $Pre, Pro, D, M$ is used.

| | Supervised | | Unsupervised | |
| --- | --- | --- | --- | --- |
| | ARI | $R^2$ | ARI | $R^2$ |
| scMoE | **1.00** | **0.61** | **0.75** | **0.61** |
| w/o DDC loss | 1.00 | 0.61 | 0.56 | 0.61 |
| w/o Recon loss | 0.97 | -0.39 | 0.60 | -1.22 |
| replace DDC w/ Cont. loss | 1.00 | 0.58 | 0.29 | 0.50 |

In Table 5, as we are targeting a multi-task setting, we also tested solely incorporating DDC loss for classification or reconstruction loss for regression in the Dyngen dataset, both in supervised and unsupervised settings. It appears that **1)** using both losses simultaneously for training benefits both tasks in the current multi-modal multi-task setting, whereas using only one fails to effectively handle the other task. Moreover, **2)** replacing DDC loss with contrastive loss is also suboptimal in the unsupervised setting, showing that DDC loss is powerful in capturing group-specific information in the multi-modal setting. Appendix D provides a comparison of computational efficiency in training time, GFLOPs, and parameters.

## 4.5 IN-DEPTH ANALYSIS OF INTERPRETABILITY

In this section, we demonstrate the interpretability of scMoE through two distinct approaches in the field of interpretable AI, i.e., mechanistic and post-hoc interpretability.

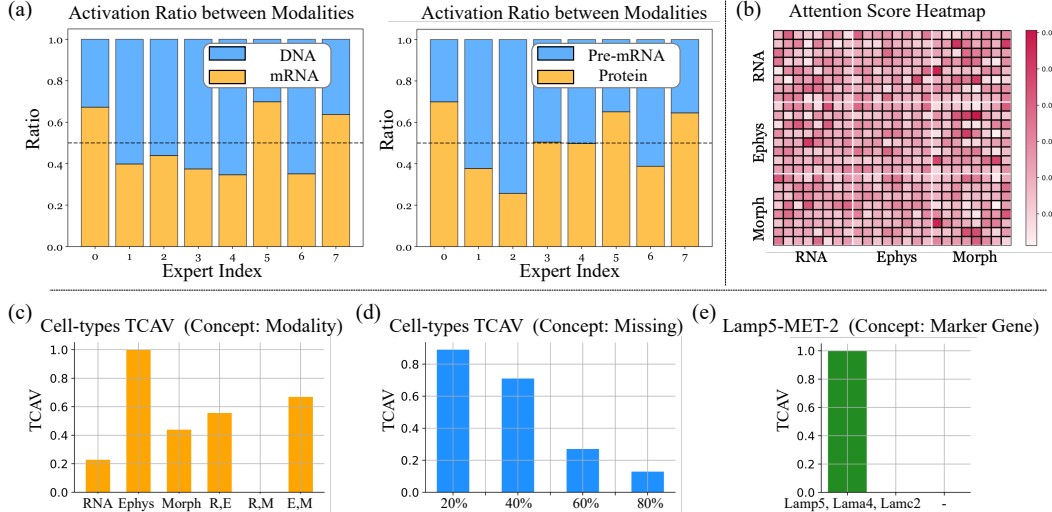

Figure 4: In-depth Analysis of Interpretability of scMoE. (a) Activation ratio between modalities in the Dyngen dataset. (b) Attention score heatmap in Patch-seq dataset. Post-hoc TCAV analysis on (c) how modality affects cell type classification, (d) how missing rate affects cell type classification, and (e) how marker genes affect rare cell type identification.

**Mechanistic Interpretability via SMoE and MHA.** A key strength of mechanistic interpretability is its ability to provide reasoning without extra training. We highlight two insights from scMoE: 1) As shown in Figure 4(a), experts specialize in different modalities during on-the-fly selection. For example, experts $0, 5$, and $7$ focus on mRNA, while others specialize in DNA. At the same time, experts $3$ and $4$ capture more general knowledge, adeptly handling both Pre-mRNA and Protein modalities. 2) Incorporating attention enables the model to capture both intra- and inter-modality relationships (Figure 4(b)). Notably, strong cross-modality interactions—such as those between Morph and Ephys—emerge in the off-diagonal regions, underscoring the importance of modeling relationships across modalities. Thus, integrating a Multi-Head Attention layer enhances mechanistic interpretability by making these inter-modal dependencies explicit in the multi-modal domain.

**Post-hoc Interpretability via TCAV.** To complement mechanistic analysis, we adapt Testing with Concept Activation Vectors (TCAV) (Kim et al., 2018a) to the single-cell domain. TCAV constructs a CAV by training a classifier to separate examples of a concept from counterexamples, and then uses directional derivatives to quantify how much the concept influences predictions. We design domain-specific concept vectors to probe modality importance, robustness under dropout, and the role of marker genes. Figure 4(c) shows that the Ephys modality strongly influences cell-type classification in Patch-seq. In scenarios of RNA dropout, involving genes with lower missing rates (20%) improves performance (Figure 4(d)). Finally, marker genes such as Lamp5, Lama4, and Lamc2 enable identification of the rare Lamp5-MET-2 cell type (2/448 samples, Figure 4(e)). These results highlight TCAV's ability to uncover biologically grounded insights and demonstrate that even rare cell types can be identified when guided by marker gene concepts.

## 5 CONCLUSION

In this work, we investigate and design multimodal multitask learning algorithms for high-dimensional single-cell data through the lens of a Sparse Mixture-of-Experts framework, introducing scMoE. Our pilot studies identify two major challenges in existing approaches: *optimization conflict* and *costly interpretability*. To address these gaps, we adapt the Sparse Mixture-of-Experts framework for single-cell data, disentangling the parameter space to mitigate gradient conflicts across modalities. Moreover, we introduce a lightweight Concept Activation Vector to enhance post-hoc interpretability in multimodal scenarios. Comprehensive validations on both simulated and real-world datasets consistently demonstrate the effectiveness of our approach with improved interpretability. In the future, we aim to extend our pipeline to systems immunology analysis by incorporating additional imaging modalities.

## ETHICS STATEMENT

Our work advances biomedical research by boosting model accuracy and interpretability, which may support personalized medicine and improved healthcare outcomes, while also requiring careful attention to ethical issues such as patient privacy and bias.

## REPRODUCIBILITY STATEMENT

We provide our code in the supplementary materials to support reproducibility. All experiments were conducted on an A100 GPU. The datasets used are publicly available.

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

## APPENDIX

## A   THE USE OF LARGE LANGUAGE MODELS (LLMs)

We use ChatGPT[3] for grammar refinement. It was employed solely to polish text written by the authors and did not contribute to research ideation.

## B   DATASETS AND PREPROCESSING

**Dyngen Simulated Dataset.**   We use Dyngen (Cannoodt et al., 2021) to simulate a four-modality dataset comprising DNA, pre-mRNA, mRNA, and protein. The simulation generates 500 cells with each modality containing 100 dimensional features. Ground truth cell-type annotations are also provided. For Dyngen's parameters, we adopt the default settings of a linear backbone model as outlined in the Dyngen tutorial, employing functions such as `backbone_linear`, `initialize_model`, and `generate_dataset`.

**Patch-seq GABAergic Neuron Dataset.**   We utilize a Patch-seq dataset from GABAergic interneurons in the mouse visual cortex. It contains 3395 neurons retained for E-T analysis and 448 for M-E-T analysis. We standardize the input matrices for each modality to normalize the mean and standard deviation of all features in each cell to 0 and 1, respectively.

**DBiT-seq Embryo Dataset.**   It includes three modalities: mRNA expression, protein expression, and niche mRNA expression from 936 spots. We normalize mRNA expression using `scanpy.pp.normalize_total` and select the top 568 differentially expressed genes. The protein expression is similarly normalized, focusing on 22 proteins, while niche modalities derive from normalized mRNA expression. For tissue region characterization, we extract ground truth labels

---

[3]https://chatgpt.com/

from the original study and evaluate the clustering performance of UnitedNet against other methods using the adjusted rand index. For cross-modality prediction, we utilize mRNA and protein expression as inputs to UnitedNet. The DBiT-seq dataset is split into a training set ($80\%$, $748$ spots) and a testing set ($20\%$, $188$ spots) for the prediction task.

**Multiome ATAC + Gene Expression BMMCs Dataset.** We analyze a multiome ATAC + gene expression dataset from BMMC tissue across 10 donors and 4 tissue sites. Following quality control and standard preprocessing procedures, we normalize the gene expression data using median normalization, log1p transform, and standardization. We select the top $4000$ most variable genes via Scanpy. For DNA accessibility data, we binarize the matrix and select the top $13,634$ most variable features, annotating DNA-accessibility peaks with ChIPseeker and `scanpy.var_names_make_unique`.

## C HYPER-PARAMETER SETTING

**Dyngen.** For the Dyngen dataset, we set the batch size to $64$, the hidden dimension to $64$, and train for 100 epochs. We represent each modality with 4 patches and use a learning rate of $1 \times 10^{-4}$. The model's architecture includes a single layer of transformer blocks with 1 attention head. In the Sparse Mixture-of-Experts (SMoE) component, we employ 16 experts, with 2 experts being activated simultaneously.

**DBiT-seq.** For the DBiT-seq dataset, we configure the following parameters: a batch size of $64$, a hidden dimension of $64$, and 100 training epochs. Each modality is represented with 8 patches, and we employ a learning rate of $1 \times 10^{-4}$. Our model architecture consists of a single layer of transformer blocks with 4 attention heads. In the Sparse Mixture-of-Experts (SMoE) component, we utilize 8 experts, with 2 experts being activated at a time.

**Patch-seq.** The Patch-seq dataset experiments are conducted with a batch size of 32 and a hidden dimension of 128, over 100 training epochs. We represent each modality with 4 patches and set the learning rate to $4 \times 10^{-3}$. The model includes one layer of transformer blocks with 4 attention heads. The SMoE component comprises 32 experts, with 2 experts activated simultaneously.

**ATAC+gene.** In the ATAC+gene dataset, we use a batch size of 16 and a hidden dimension of 64, training for a total of 100 epochs. Modalities are represented with 8 patches each, and the learning rate is set to $1 \times 10^{-4}$. The architecture includes a single layer of transformer blocks with 1 attention head. We employ 8 experts in the SMoE component, with 2 experts being activated per token.

## D COMPUTATIONAL EFFICIENCY

Table 6: Training Time (s) $\downarrow$

|  | Modality Combiations | | | |
| --- | --- | --- | --- | --- |
|  | Pre, m | Pro, m | D, m | Pre, Pro, D, m |
| scMoE | **26.83** | **27.48** | **27.53** | **45.93** |
| UnitedNet | 34.21 | 34.23 | 36.01 | 61.02 |

Table 7: GFLOPs $\downarrow$

|  | Modality Combiations | | | |
| --- | --- | --- | --- | --- |
|  | Pre, m | Pro, m | D, m | Pre, Pro, D, m |
| scMoE | **0.01** | **0.01** | **0.01** | **0.03** |
| UnitedNet | 0.10 | 0.10 | 0.10 | 0.25 |

We evaluated the computational efficiency of scMoE compared to the recent state-of-the-art model, UnitedNet, using the Dyngen dataset across various modality combinations. In terms of time (Ta-

Table 8: Parameters (K) $\downarrow$

|  | Modality Combiations | | | |
|---|---|---|---|---|
|  | Pre, m | Pro, m | D, m | Pre, Pro, D, m |
| scMoE | **0.05** | **0.05** | **0.05** | **0.08** |
| UnitedNet | 1.39 | 1.39 | 1.39 | 2.74 |

ble6), GFLOPs (Table 7), and number of parameters (Table 8), we observe notable gains in computational efficiency, primarily due to the adoption of the SMoE layer, which selectively activates only the most relevant experts for a given input token. In contrast, UnitedNet does not employ sparse activation in its fusion layer, leading to suboptimal computational efficiency and overall performance in downstream tasks due to gradient conflicts in a multi-modal, multi-task scenario. Specifically, scMoE achieves up to 24.7% faster training time, $8.3\times$ lower GFLOPs, and a $34.3\times$ reduction in parameters, demonstrating its scalability and efficiency. These results highlight the advantage of sparse activation, allowing our model to achieve competitive performance while significantly reducing computational costs.

