# OpenReview forum: "scMoE: single-cell Multi-Modal Multi-Task Learning via Sparse Mixture-of-Experts"
_ICLR.cc/2026/Conference — Submitted to ICLR 2026_

### Official Review · Reviewer_3MBe · 2025-10-27

**Soundness:** 3
**Presentation:** 3
**Contribution:** 2
**Rating:** 6
**Confidence:** 5

**Summary:**

In this work, the authors propose scMoE, a novel framework that integrates a Sparse Mixture-of-Experts (SMoE) layer into a transformer architecture for single-cell multi-modal multi-task learning. The method aims to address two identified challenges in existing approaches: optimization conflicts arising from gradient interference in shared parameter spaces, and the high computational cost associated with post-hoc interpretability methods like SHAP. Through extensive experiments on simulated (Dyngen) and real-world datasets (DBiT-seq, Patch-seq, ATAC+gene), the authors demonstrate that scMoE achieves superior or competitive performance in joint group identification and cross-modal prediction tasks compared to several state-of-the-art baselines. A notable advantage of scMoE is its enhanced computational efficiency and inherent mechanistic interpretability, afforded by the sparse activation of experts and the accompanying gating network. The authors further augment interpretability by adapting Concept Activation Vectors (TCAV) to the single-cell domain. While the application of SMoE to this specific problem is novel and well-executed, the core architectural innovation primarily involves the adept adaptation and integration of existing components (Transformer, SMoE) from other domains.

**Strengths:**

This paper presents a solid contribution to the field of single-cell data analysis, with several strengths that merit consideration for acceptance:

**1. Well-Motivated Problem Formulation:** The paper clearly identifies and articulates two significant, yet under-addressed, challenges in multi-modal single-cell analysis: the optimization conflict in multi-task learning and the impractical computational cost of prevalent post-hoc interpretability methods. This provides a compelling rationale for the proposed research.

**2. Comprehensive Empirical Validation:** The authors provide thorough experimental evidence across one simulated and three real-world benchmark datasets. The results consistently show that scMoE outperforms strong baselines in both joint group identification and cross-modal prediction tasks, lending substantial support to the method's efficacy and generalizability.

**3. Enhanced Computational Efficiency and Interpretability:** A key advantage of scMoE is its dual benefit of significantly improved computational efficiency (in training time, GFLOPs, and parameters) due to sparse activation, and the provision of inherent, mechanistic interpretability through the gating network and attention mechanisms, moving beyond costly post-hoc explanations.

**Weaknesses:**

While the paper presents a commendable approach, several limitations and unresolved questions temper the enthusiasm for its groundbreaking impact, positioning it as a solid incremental contribution rather than a transformative one.

**1. Limited Novelty in Core Mechanism:** The core innovation lies in the strategic application and integration of the SMoE architecture to a new domain, rather than in a fundamental algorithmic advancement. SMoE is a well-established technique in ML, and the transformer architecture is ubiquitous. The paper would be strengthened by a more nuanced discussion of what specific architectural adaptations were necessary for single-cell data beyond the patching technique, which is itself borrowed from Vision Transformers.

**2. Narrow Scope of Multi-task Learning Evaluation:** The paper evaluates a specific pair of tasks (joint clustering and cross-modal prediction). However, the broader promise of multi-task learning often includes positive transfer and robustness. The work does not explore whether the scMoE framework facilitates knowledge transfer between tasks more effectively than baselines, or if it improves robustness to noise or missing modalities in a more challenging setting, which would be a stronger validation of its multi-task capabilities.

**Questions:**

1. A well-known challenge in MoE models is load imbalance, where a few experts dominate the computation. Common solutions employ an auxiliary load balancing loss (e.g., Shazeer et al., 2017; Lepikhin et al., 2021). However, there doesn't seem to be any related design in scMoE. Could the authors clarify whether this issue was encountered in scMoE? If so, what specific mechanisms are implemented to ensure balanced utilization of experts? If not, why?

2.  In Appendix C, the model configurations (e.g., number of attention heads, experts) are heavily customized for each dataset. This raises a concern regarding the generalizability and practical utility of scMoE. If it aims to be a foundational model for single-cell data, why does its performance seem to depend so critically on dataset-specific tuning? Furthermore, for practitioners who wish to apply scMoE to a new dataset, what principled guidelines or methodology does the authors propose for determining the optimal model architecture, rather than relying on exhaustive search?

3. Figure 3(a) evaluates models on a tissue region identification task using the DBiT-seq dataset, which inherently contains spatial information. However, the methodology section does not describe any mechanism for scMoE to incorporate spatial coordinates as input. Could the authors clarify what specific features were used as input for this task? If only gene expression and/or protein data were used without spatial context, the experimental setup may be misaligned with the intrinsic objective of spatial transcriptomics analysis, and the comparison with other methods in this context would be difficult to interpret fairly. Please justify the design of this experiment.

---

### Official Review · Reviewer_GnLx · 2025-10-30

**Soundness:** 3
**Presentation:** 1
**Contribution:** 2
**Rating:** 2
**Confidence:** 4

**Summary:**

The paper presents a framework named scMoE that incorporates a Sparse Mixture-of-Experts (SMoE) mechanism into a transformer-based architecture for single-cell multi-omics analysis. The method includes encoders for embedding multi-modal data, a transformer block with SMoE and cross-attention for learning intra- and inter-modal relationships, and decoders for reconstruction. It aims to address optimization conflicts and interpretability challenges in multi-modal learning. The framework is evaluated on simulated and multiple real-world single-cell datasets, focusing on joint group identification and cross-modal prediction tasks.

**Strengths:**

- The paper provides a clearly structured methodology with a well-defined motivation for the proposed approach.
- It presents a design applicable to various tasks in single-cell multi-modal data analysis.
- The experimental evaluation includes both simulated and real-world datasets, illustrating the framework’s applicability and consistency across different types of multi-modal single-cell data.

**Weaknesses:**

- The paper’s layout is suboptimal. The introduction of the term “concept-activation vectors” feels abrupt. To improve readability, it would be helpful to provide a brief explanation for this term in the introduction.

- The figures and tables are inadequately presented. Specifically, the definition and calculation procedure of the ARI in Figure 1 and Tables 1 and 2 are unclear. Additionally, the horizontal and vertical axes in Figure 1(b) are not clearly labeled, and the two figures in Figure 4(a) lack sufficient explanation.

-  Hyperparameter tuning is not clearly justified. In Appendix C, the authors provide hyperparameter details for scMoE, but it appears that different settings were used for each dataset. The author did not explain the reason for choosing these settings.

- The claimed improvement in interpretability with scMoE is not convincingly demonstrated. Expert activations or multi-headed attention mechanisms provide limited new insights, and the TCAV method could be applied to any baseline model. The paper does not clearly show what new biological or methodological insights are gained through scMoE that were not accessible with previous workflows.

- The proposed post-hoc interpretability method based on Concept Activation Vectors is only briefly mentioned and lacks extensive validation or comparison with other post-hoc interpretability techniques.

-  Other concerns: Regarding the ablation study, the main text incorrectly refers to Table 4.4, which does not exist; it should instead refer to Table 4. Furthermore, from Table 4, it is not clear why selecting the top-2 from 16 experts is considered optimal, since Table 4 does not include results for N=16 and K=2. If the corresponding results exist elsewhere, the authors should include them in Table 4 to improve readability. Additionally, if selecting 2 out of 16 experts is indeed the best choice, it is unclear why the four datasets were not handled in the same way.

- Why does scMoE fail to achieve the best performance across the four modalities? This appears to weaken the motivation for scMoE as a solution to optimization conflicts.

**Questions:**

See in the weakness section above.

---

### Official Review · Reviewer_eF2C · 2025-10-31

**Soundness:** 2
**Presentation:** 3
**Contribution:** 2
**Rating:** 2
**Confidence:** 4

**Summary:**

Multi-modal learning has seen recent advances due to the growth in multi-omics data. However, multi-task learning of omics modalities is bottlenecked by (1) optimization conflict and (2) costrly interpretability. In the case of former, irregular gradients across modalities deteriorate learning. As for latter, current interpretability methods rely on post-hoc analysis. The paper addresses these challenged by proposing scMoE, a framework that incroporates sparse MoE within the transformer block for learning multi-omics single-cell representations. Per-modality encoders and decoders are utilized and trained using DDC and variational inference objectives. The model provides interpretability by means of analyzing expert heads and cross-attention patterns. Experiments demonstrate effectiveness across modality combinations and real-world tasks of tissue group ientification and cell types classification.

**Strengths:**

* The paper is well written and organized.
* The paper presents an intuitive perspective on multi-omic learning.

**Weaknesses:**

* **Motivation & Central Problem:** The paper motivates the use of sparse MoE architecture using optimization conflict and costly interpretability. However, the paper does not validate these problems. Authors do not explain how MoEs can address these issues and provide empirical evidence for the same. Does scMoE provide better gradient synchronization across modalities? How does gradient stability compare across different models and modality configurations? How does scMoE make the model more interpetable compared to traditional methods like SHAP? In its current form, majority of experts capture local features which can be observed from modality ratios. These patterns do not explain relationship between modalities or the effectiveness of MoE in constructing better representations.

* **Contribution & New Knowledge:** I am struggling to understand the effectiveness of contribution and new knowledge added by the paper. Note that this is not looking at the novelty of the work. Paper does not provide the reader with new knowledge such as the utility or inner workings of the MoE module. In its current form, the work suggests a trivial application of sparse MoEs to the transformer block for learning multi-modal multi-omics representations. However, this does not throw light on or explain the design decisions considered in the work. Authors could consider comparing between sparse and soft MoEs, study the space of experts and routing mechanism in the case of different modality combinations in detail.

* **Datasets:** The paper considers a range of datasets across multiple modalities. However, the setting is fairly limited in its proportion and more tailored towards the breadth of data. That is, each dataset has only a handful of samples (except for ATAC+gene) but a larger number of modalities which is not ideal for studying MoE architectures. Generally, MoEs are found to be perfoermant when the data per expert and hence the diversity in samples is large. How do experts adapt to the breadth of datasets? Does each expert learn different disentangled features across modalities? Results in figure 4 show that experts have overlapping activations for madlity pairings.

* **Interpretability:** The paper attempts to understand and interpret the working of MoEs but does not do so in a mechanistic manner. At its core, MoEs are used as black-box approximators and ratios do not provide intuitive insights into their operations. Attention patterns highlight inter and modal relationships but it is nontrivial to understand cross-modal patterns at the granular level. Authors should consider taking a deeper look at the MoE and its interpretability. How are MoE weights distributed? Do the top $n$ weights belong to a particular expert? Which experts are activated corresponding to different modality ablations? Do experts capture modality-specific information? Does overfitting to a modality cause similar expert selection from the router?

**Questions:**

Refer to weaknesses

---

### Meta-Review · Area_Chair_xCvz · 2026-01-06

**Summary:**

This paper studies an interesting problem, but the proposed model does not offer enough new ideas, and its explanations are too limited to support proper validation. These issues led two reviewers to give negative evaluations, and no rebuttal was provided to address their concerns. Although the paper claims that sparse MoE models help with optimization and interpretability, it does not clearly explain or demonstrate how this actually happens. The analysis of expert behavior is shallow and mostly descriptive, making it hard to understand what the model learns or why it works.

In addition, the paper does not clearly communicate what new insights it contributes beyond a straightforward use of existing MoE techniques. The datasets used are small and not well suited for testing MoE models, which typically require large and diverse data to show clear benefits. As a result, the experiments do not convincingly show that different experts learn meaningful or distinct information.

**Reviewer Concerns:**

NO rebuttal.

**Reviewer Scores:**

No idea without rebuttal.

---

### Decision · Program_Chairs · 2026-01-26

Reject